# Low-Cost Orientation Determination System for CubeSat Based Solely on Solar and Magnetic Sensors

**DOI:** 10.3390/s23146388

**Published:** 2023-07-14

**Authors:** Yerkebulan Nurgizat, Abu-Alim Ayazbay, Dimitri Galayko, Gani Balbayev, Kuanysh Alipbayev

**Affiliations:** 1Institute of Telecommunications and Space Engineering, Almaty University of Power Engineering and Telecommunications Named G. Daukeev, Almaty 050062, Kazakhstan; work_abu@hotmail.com (A.-A.A.); k.alipbayev@aues.kz (K.A.); 2LIP6 Laboratory, Sorbonne University, 75004 Paris, France; dimitri.galayko@sorbonne-universite.fr; 3Academy of Logistics and Transport, Almaty 050062, Kazakhstan; g.balbayev@gmail.com

**Keywords:** CubeSat, satellite orientation, solar sensors, magnetic sensors, attitude determination, satellite–Sun vector, magnetic field orientation, quaternions

## Abstract

CubeSats require accurate determination of their orientation relative to the Sun, Earth, and other celestial bodies to operate successfully and collect scientific data. This paper presents an orientation system based on solar and magnetic sensors that offers a cost-effective and reliable solution for CubeSat navigation. Solar sensors analyze the illumination on each face to measure the satellite’s orientation relative to the Sun, while magnetic sensors determine the Earth’s magnetic field vector in the satellite’s reference frame. By combining the measured data with the known ephemeris of the satellite, the satellite–Sun vector and the magnetic field orientation can be reconstructed. The orientation is expressed using quaternions, representing the rotation from the internal reference system of the satellite to the selected reference system. The proposed system demonstrates the ability to accurately determine the orientation of a CubeSat using only two sensors, making it suitable for installations where more complex and expensive instruments are impractical. Additionally, the paper presents a mathematical model of a low-cost CubeSat orientation system and a hardware implementation of the sensor. The technology, using solar and magnetic sensors, provides a reliable and affordable solution for CubeSat navigation, supporting the increasing sophistication of miniature payloads and enabling accurate satellite positioning in space missions.

## 1. Introduction

An important aspect of CubeSat operations is determining the satellite’s orientation relative to the Sun, Earth, and other celestial bodies. The technology used in the development of CubeSat orientation sensors includes solar and magnetic sensors [1,2]. This information is necessary for spacecraft orientation [3,4], control, and collecting accurate scientific data [5]. The purpose of the solar sensor is to determine the coordinates of the satellite–Sun vector [6,7] in the reference frame associated with the satellite. This is achieved by processing photoelectric signals from solar sensors located on the six sides of the satellite [8,9].

Modeling shows that the precision of attitude determination can be better than 1°. Furthermore, the Radio Aurora Explorer 3U CubeSat [10] demonstrated, back in 2014, that it is possible to obtain information about the satellite’s position with accuracy better than 1° (solar light) in orbit [11,12]. This is especially true when the satellite is on the sunny side, as multiple sensors are used for successful task execution. However, this determination is more difficult during eclipses [13]. Determining the orientation of CubeSat satellites remains a challenging task, as they are small and still lack active orientation and control systems [14,15]. Additionally, new miniature payloads for space-based missions are becoming increasingly complex and require accurate knowledge of a satellite’s position. 

One of the main goals of the SAT mission is to develop a CubeSat orientation algorithm that will exceed a few degrees of accuracy when using two sensors, particularly in sunlight. This objective aims to overcome the limitations of using complex and expensive tools like star sensors. Consequently, this paper presents an orientation system based solely on two sensors: a solar sensor and a magnetometer, which can be easily installed on any CubeSat. This system offers a practical alternative for situations where more elaborate instruments are not feasible.

Still, the utilization of solar and magnetic sensors in CubeSat orientation systems provides several advantages. Firstly, it is a cost-effective and reliable solution, making it accessible for a wide range of CubeSat missions. Secondly, the availability of solar and Earth’s magnetic fields throughout most regions on our planet enables the utilization of this technology for CubeSat navigation, regardless of the mission’s location.

In summary, this paper aims to present a comprehensive orientation system for CubeSats based on solar and magnetic sensors, highlighting its feasibility and advantages. The proposed system, utilizing only two sensors, offers a practical solution for determining CubeSat orientation and addresses the challenges posed by limited hardware and energy resources. Through its cost-effectiveness, reliability, and accessibility, the technology opens up new possibilities for CubeSat missions, supporting their growing complexity and enabling precise knowledge of the satellite’s position in space-based operations.

The scope of this orientation system based on solar and magnetic sensors extends to CubeSat operations and scientific data collection, and also encompasses the development of a mathematical model, hardware implementation, and practical application of the orientation system.

The novelty of this work lies in the mathematical model, which is a new methodology for more accurate determination of the orientation of nanosatellites using budget sensors. 

The article is structured as follows: Section 2 describes the mathematical model of the satellite orientation system and presents the sensor architecture. Section 3 discusses the operational algorithm and the hardware implementation of the sensor. In Section 4, a detailed presentation of the results obtained from studying the algorithm of the proposed system in laboratory conditions is provided. The concluding remarks are presented in Section 5.

## 2. Materials and Methods

This chapter presents a mathematical model of the orientation system and provides detailed descriptions of the hardware architecture of the sensors.

### 2.1. Theory of the Proposed Attitude Sensor

In this subchapter, a mathematical model based on the solar and magnetic sensors of the passive orientation system of the nanosatellite is constructed.

The solar and magnetic sensors are used to collect raw information about the CubeSat’s orientation: the satellite–Sun vector and the magnetic field orientation. If the satellite’s position in orbit is known, these data allow for the recovery of the satellite’s attitude.

Solar sensors measure the orientation of the satellite relative to the Sun. These sensors measure the illumination of each face of the satellite, and the processing unit computes the direction of the satellite–Sun vector *S* [16,17]. Magnetic sensors are used to measure the Earth’s magnetic field vector B in the satellite’s reference frame.

If the ephemerides of the satellite are known, it is possible to determine the reference values of the magnetic field vector and the satellite–Sun vector from publicly available astronomical databases and models. By measuring the positions of the Sun and the value of the magnetic field using the satellite’s sensors at a specific moment in time and comparing these data with real data, we obtain information about the satellite’s position in the selected reference system. The attitude of a satellite can be conveniently expressed using quaternions. A quaternion is a mathematical object defined by four real numbers that represent a rotation in a three-dimensional space. The satellite’s attitude is defined by a quaternion that rotates the internal satellite reference frame to align it with a selected reference frame (e.g., an inertial geocentric frame). In this work, the selected reference frame for attitude expression is J2000.

Both reference vectors *B* and *S* are expressed in the J2000 geocentric inertial coordinate system and denoted as *Bj* and *Sj*. The satellite’s sensors measure these same vectors in coordinates related to the satellite and denoted as *Bs* and *Ss*. The position of the satellite is determined by a quaternion that aligns the satellite’s coordinate system with the J2000 system. This quaternion is defined by the rotation that aligns two pairs of vectors: *Bs*, *Ss*, and *Bj*, *Sj*. This rotation is calculated in two steps. In the first step, a rotation that aligns the vectors *Bs* and *Bj* calculated by determining the direction vector of the rotation axis using Equation (1). This vector is equal to the normal vector to the plane formed by vectors *Bs* and *Bj.*
(1)$$\vec{n}=\left[\vec{Bj}\times \vec{Bs}\right]/\left(\frac{\vec{Bj},\;\;\;\vec{Bs}}{\left|\vec{Bj}\right|,\;\;\left|\vec{Bs}\right|}\right)$$

Then, the angle between vectors *Bs* and *Bj* is calculated using Equation (2):(2)$$\vec{y}=\mathrm{acos}\left(\left|\vec{Bs}\right|\times \left|\vec{Bj}\right|\times \cos\alpha \right)/\left(\frac{\vec{Bs}}{\left|\vec{Bs}\right|}/\frac{\vec{Bj}}{\left|\vec{Bj}\right|}\right)$$

And a quaternion for the first rotation can be expessed in form:(3)$$q1=\left[cos\left(y/2\right)\;\vec{n}*sin\left(y/2\right)\right].$$

During this rotation, vector *Bs* is transformed into vector *Bj*, while vector *Ss* takes a new value *Ss*′ Equation (4):(4)$$Ss^{\prime }=\left[\begin{array}{ccc} 1-2q1_{2}^{2}-2q1_{3}^{2} & 2(q1_{1}q1_{2}+q1_{0}q1_{3}) & 2\left(q1_{1}q1_{3}-q1_{0}q1_{2}\right)\\ 2\left(q1_{1}q1_{2}-q1_{0}q1_{3}\right) & 1-2q1_{1}^{2}-2q1_{3}^{2} & 2(q1_{2}q1_{3}+q1_{0}q1_{1})\\ 2(q1_{1}q1_{3}+q1_{0}q1_{2}) & 2(q1_{2}q1_{3}-q1_{0}q1_{1}) & 1-2q1_{1}^{2}-2q1_{2}^{2} \end{array}\right]\left[\begin{array}{c} Ss_{1}\\ Ss_{2}\\ Ss_{3} \end{array}\right]$$

After the first rotation, vector *Bs* is transformed into vector *Bj*, and vector *Ss* takes on a new value, *Ss*′.

In the second step, it becomes necessary to rotate vector *Ss*′ around the axis defined by vector *Bj* in order to align it with vector *Sj*. However, achieving perfect alignment is challenging due to measurement errors. The angles between the initial vectors *Bs*, *Ss*, *Bj*, and *Sj* are likely to differ, although they should ideally be the same in theory.

Therefore, a rotation of vector *Bj* is calculated to minimize the difference between the rotated vector *Ss*′ and vector *Sj*’s norm. An optimization algorithm, such as the gradient descent method, can be employed to find the rotation angle that minimizes this difference. This is accomplished by defining a function *f*(α) that calculates the Euclidean difference between the rotated vector, *Ss*′, and vector *Sj*. The rotation is determined by the rotation axis, which is equal to *Bj*, and the angle α, using Equation (5):(5)$$f\left(a\right)=\left|\left|Sj-rot\left(Sj,\;norm\left(Bj\right),\;a\right)\right|\right|,$$
where, *rot* (x,n,α) is the result of rotating the vector x by angle α around the axis n. Using the optimization algorithm, α is found that minimizes the value of the function. 

Now it is possible to calculate the second rotation quaternion, ‘q2’, using Equation (6):(6)$$q2=\left[cos\left(alpha/2\right)\;\vec{Bj}*sin\left(alpha/2\right)\right].$$

Finally, the quaternion expressing the orientation of the satellite is obtained by multiplying quaternions *q*1 and *q*2 using Equation (7):(7)q=q1∗q2

The result is verified in two steps. First, one calculates the triple product of vectors using Equation (8):(8)$$dot\left(cross\left(quatrotate\left(q,\;Ss\right),\;Bj\right),\;Sj\right)$$
where, *dot* is a dot (x,y) product operator, cross (x,y) is a cross product operator, quatrotate (q,x) is a function rotating a vector x according to the quaternion q. If the calculation is correct, the result should be close to zero (it means that the aligned vectors and vector *Sj* are on the same plane).

The second verification step consists of comparing the vectors *Bj* and the rotated vector *Bs* using Equation (9):(9)$$\left|\left|Bj-quatrotate\left(q,\;Bs\right)\right|\right|$$

The result should also be close to zero.

### 2.2. Hardware Architecture

In this subchapter, the hardware architecture of the sensor connection to the microcontroller is implemented.

An orientation determination system for CubeSats based on solar and magnetic sensors can be implemented using the following hardware architecture, which is shown in Figure 1:

Six solar sensors, such as photodiodes, were placed on each face of the CubeSat to measure the intensity of sunlight. The 6 faces of the CubeSat were determined as follows +x, −x, +y, −y, +z, −z.

A three-axis magnetometer was used to measure the Earth’s magnetic field. The magnetometer was placed in a suitable location on the CubeSat to measure the intensity and direction of the magnetic field.

A microcontroller was used to interact with the solar sensors and magnetometer for data processing. It was programmed to perform calculations to determine the satellite’s orientation.

The overall system architecture is relatively simple and inexpensive as it only requires a few inexpensive sensors, a microcontroller, and power source components. The system can be integrated into a CubeSat during the design phase or added as an upgrade to existing CubeSat equipment.

### 2.3. Architecture and Model of Magnetic Sensor

In this subchapter, the architecture of the magnetic sensor used is presented.

Electronic three-axis magnetometers based on anisotropic magneto-resistive (AMR) technology are used as magnetic sensors on CubeSats to measure the magnetic field strength in different directions. The magnetometer used is the LSM303DLH, an inexpensive three-axis digital magnetometer capable of measuring magnetic fields in the range of ±8 Gauss [18]. The LSM303DLH magnetic sensor is a digital compass that measures the Earth’s magnetic field in three dimensions (X, Y, Z). The sensor is located such that its own coordinate axes coincide with the axes of the satellite’s internal coordinate system, thus directly outputting the value of the *Bs* vector.

## 3. Practical Implementation of the Sensor Relative Attitude

In this chapter, a practical application of the orientation system is demonstrated, and the electrical circuit for sensor connections is presented. Additionally, a practical implementation in the form of a ready-made board is provided.

For the implementation of the above-proposed system, we used Arduino. The mathematical model of the CubeSat’s orientation system was transformed into software code in the Arduino IDE. The block diagram of the system is shown in Figure 2.

The orientation system computation algorithm consists of several main steps:-Reading variables from solar panels. For more accurate data, the program reads 150 values from each panel and outputs the average value. Then, the data from the solar panels are normalized.-Reading variables from the magnetic sensor and normalizing them.-Calculating the quaternion for the first rotation.-Calculating the quaternion for the second rotation.-Calculating the quaternion that expresses the overall orientation of the CubeSat.

Based on the constructed architecture of the CubeSat orientation system, a connection diagram of the sensors to the microcontroller was developed, as well as the manufacture of the orientation system board. The electrical circuit diagram of the system is shown in Figure 3.

Control of the actuating devices is carried out using the Atmega2560 (Microchip Technology Inc., Chandler, AZ, USA) microcontroller (U1). To operate the microcontroller at a clock frequency of 16 MHz, an HC-49S quartz resonator is used.

To connect the solar panels to the microcontroller (U1.1), current-limiting resistors of 86 kOhm and ports from A0 to A5 are used. The signals from the solar panels are fed to the microcontroller’s 10-bit ADC for further processing.

The magnetometer is connected to the microcontroller via the I2C serial asymmetric bus, through port 20 and 21. The processed data is written to an SD card, which is connected to the microcontroller via the Serial Peripheral Interface (SPI).

The device is powered by an AMS1117 5 V linear voltage regulator, which converts the 12 V DC voltage to 5 V DC voltage, corresponding to the power supply level of the microcontroller’s peripherals and external data-storage devices. As an alternative, a backup power source, such as a small battery, can be connected to ensure the system continues to function during power outages.

The radio electronic circuit of the device was designed using the EasyEda STD software environment, and the printed circuit board for the system was manufactured using a CNC milling machine (vmc650, Shandong, China). Files for cutting the foil-textolite were prepared in the flatcam program, as shown in Figure 4 and Figure 5.

## 4. Experimental Results

This chapter presents the results of experiments conducted to test the proposed passive orientation system of the nanosatellite in the laboratory conditions of the university.

The experiment was conducted in the university’s laboratory. An electric lamp was chosen as a Sun emulator, and a supporting frame was used to install the CubeSat, which is shown in Figure 6.

Before characterizing the developed orientation sensor, calibration of the measurement instruments was carried out. Using an online calculator, the data of the Earth’s position in the heliocentric reference system for the year 24 April 2023 were calculated [19], as well as the magnetic field data in the geocentric reference frame at the laboratory location [20].

The experiment was conducted in two stages.

The first stage of the experiment was carried out using solar sensors. The satellite was mounted on a three-axis holder at a distance of 32 cm from the solar emulator. The goal of the experiment was to determine the satellite–Sun vector and compare it with the real satellite–Sun orientation. As part of the experiment, solar panels were illuminated at different angles, and the following data were obtained:
(a)First, one side of the CubeSat was illuminated. In Figure 7 below, it can be seen that the angle of incidence of the beam on the y+ panel is 90° degrees. The table on the left shows a sample of the data output by sequential measurements (a total of 150 for determining each value). Each value corresponds to the projection of the Sun-satellite vector on the normal vector of the corresponding panel. These values correspond to the component of the guiding vector satellite–Sun, obtained by normalizing the illumination value measured for the panel. For panels that are in shadow, this value is always zero. Thus, the value of the vector, satellite–Sun, is always determined by the sum, $\underset{i}{\sum }\alpha_{i}\vec{n_{i}}$, where α_i_—is the normalized illumination reading of panel i, and $\underset{i}{\sum }\vec{n_{i}}$—is the normal vector of the panel. The displayed data in Figure 8 show that $\vec{n_{ns}}=\vec{n_{y+}}$, where $\vec{n_{y+}}$—is the normal vector of the y+ panel.(b)Next, the lighting was set up so that sunlight falls on both sides of the CubeSat. Two different measurements were taken:When the Sun’s rays fall equally on both panels (angle of incidence 45°, Figure 8a).When the two panels are illuminated unevenly, Figure 8b.


In both cases, the sensor accurately determines the angle of incidence of the Sun’s rays, and using this data, we found the coordinates of the vector $\vec{n_{ns}}$. For example, in the case shown in Figure 8b, we have $\vec{n_{ns}}=0,86\vec{n_{x+}}+0,51\vec{n_{y+}}$.

(c)In the following experiment, the Sun illuminates three faces of the satellite. Two experiments were carried out: one when the Sun’s ray illuminates three panels equally (angle of incidence α = 60.8°, (Figure 9a), and the second case when one of the panels is illuminated at different angles (Figure 9b). In both cases, the data from the solar sensors are correct. For example, in the second experiment, we have $\vec{n_{ns}}=0,67\vec{n_{z+}}+0,67\vec{n_{x+}}+0,32\vec{n_{y-}}$.

The second stage of the experiment included both Sun sensors and a magnetic sensor. To implement the experiment, we chose a reference frame associated with the laboratory room, replacing J2000 for our laboratory experiment. The Sun emulator was placed along the *X*-axis, so the vector *Sj* was equal to:$$Sj=\left[1\;0\;0\right];$$

In this coordinate system, the IGRF model gives the following value for the magnetic field:$$Bj=\left[-0.251\;\;-0.004\;\;0.996\right]\;\left[µT\right];$$

After loading the coordinates and installing the satellite, we rotated the satellite relative to the Sun with a 20° step and recorded the results of the satellite sensors along the way.

(a) During the first experiment, the CubeSat rotated around an axis passing through the center of the faces z+ and z−. Accordingly, the Sun’s rays fell in such a way that they illuminated two panels simultaneously (+x, +y) as shown in Figure 10. The magnetic sensor was installed in such a way that its own axes coincided with the axes of the satellite, aligning with the normal directions to the solar panels. Subsequently, data were collected from the magnetic and solar sensors, from which orientation information was extracted. Additionally, prior to the experiment, we calculated the theoretical quaternion for each angle. These data are presented in Table 1. 

After the first experiment, we need to calculate the average error for each component. To calculate the average error between two quaternions, you can use the formula for the average absolute error given by Equation (10):(10)$$Average\;error=\left(\left|q1-p1\left|+\right|q2-p2\left|+\right|q3-p3\left|+\right|q4-p4\right|\right)/4,$$

Here, *q*1, *q*2, *q*3, *q*4 are the components of the theoretical quaternion, and *p*1, *p*2, *p*3, *p*4 are the components of the measured quaternion.

Analyzing the results of the first experiment (Figure 11), we apply the formula for the average absolute error and draw the following conclusion: an error of 1.2% has been identified, indicating a substantial and satisfactory agreement between the working hypothesis, theoretical assumptions, and the experimental results.

Experiment 2: The objective of the second experiment was to illuminate three faces of the CubeSat simultaneously. The experiment was conducted as follows:

The satellite was positioned such that the Sun equally illuminated three panels adjacent to one of the vertices (Figure 12). Simultaneously, a magnetic sensor was installed inside the satellite, aligned with the local magnetic axis as the initial position. Illumination readings of the satellite panels were then taken. The experimental results corresponding to the satellite’s vertices are shown in Table 2.

By comparing the analysis from the tables, it can be observed that the Sun’s rays fell on the three satellite panels in a highly consistent manner. Additionally, this algorithm calculates the orientation of the nanosatellite, specifically the quaternion.

## 5. Conclusions

This paper includes a discussion of the results obtained from laboratory tests conducted on an inexpensive CubeSat in the CubeSat format. The primary objective of these tests was to determine the coordinates of the satellite–Sun vector in the satellite’s reference frame, using magnetic sensors and Sun sensors. These coordinates were then utilized to accurately determine the CubeSat’s orientation. By incorporating a magnetic sensor and developing a mathematical model of the CubeSat’s passive orientation system, experimental testing was conducted under controlled laboratory conditions. The results of these tests demonstrated the system’s capability to accurately determine the satellite’s orientation throughout the orbital day.

In the first experiment, the proposed sensor of the passive orientation system of the nanosatellite was tested. A comprehensive analysis of the solar panels was performed, and a mathematical model for the solar sensors was constructed. Additionally, experiments were conducted at various distances from the solar simulator. The detailed data and findings of this experiment can be found in the source provided [16].

In the second experiment, aimed at enhancing the accuracy of the nanosatellite’s orientation, a second sensor was incorporated to measure the Earth’s magnetic field. Following the integration of this second sensor, a mathematical model of the complete passive orientation system was developed. The outcome of this experiment yielded an orientation sensor with an error rate of 1.2%, which was a significant achievement in laboratory conditions.

In conclusion, the exclusive utilization of solar and magnetic sensors for orientation determination presents a cost-effective solution for CubeSats. These sensors are generally affordable and have low power requirements, making them particularly advantageous for CubeSats with limited resources. Moreover, the combination of information gathered from these sensors enables accurate orientation determination without the need for additional sensors or complex algorithms. This straightforward and cost-effective method of orientation determination is a noteworthy advancement in enhancing access to spaceflight for a broader range of organizations.

It is worth noting that this system exhibits potential for further expansion in the future. It can be adapted to accommodate various installations, incorporate various sensor types, facilitate scientific research, and investigate various anomalies. By harnessing the benefits of solar and magnetic sensors, CubeSat missions can continue to evolve and effectively address the increasing demands of space exploration and scientific investigations.

In summary, the results obtained from the laboratory tests validate the effectiveness and reliability of the proposed orientation system based on solar and magnetic sensors. These results instill confidence in the system’s ability to accurately determine the CubeSat’s orientation, thereby supporting the successful operation and data collection of CubeSat missions. With its affordability, simplicity, and adaptability, this orientation system offers a promising avenue for advancements in CubeSat technology and expands the opportunities for organizations to participate in space-based activities.

## Figures and Tables

**Figure 1 sensors-23-06388-f001:**
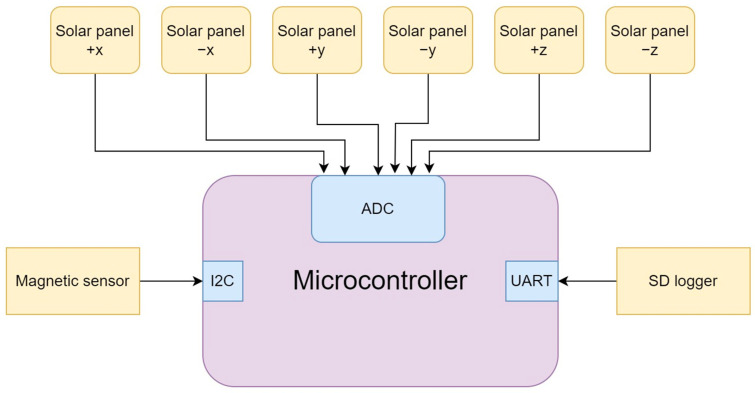
Architecture of the proposed CubeSat orientation system.

**Figure 2 sensors-23-06388-f002:**
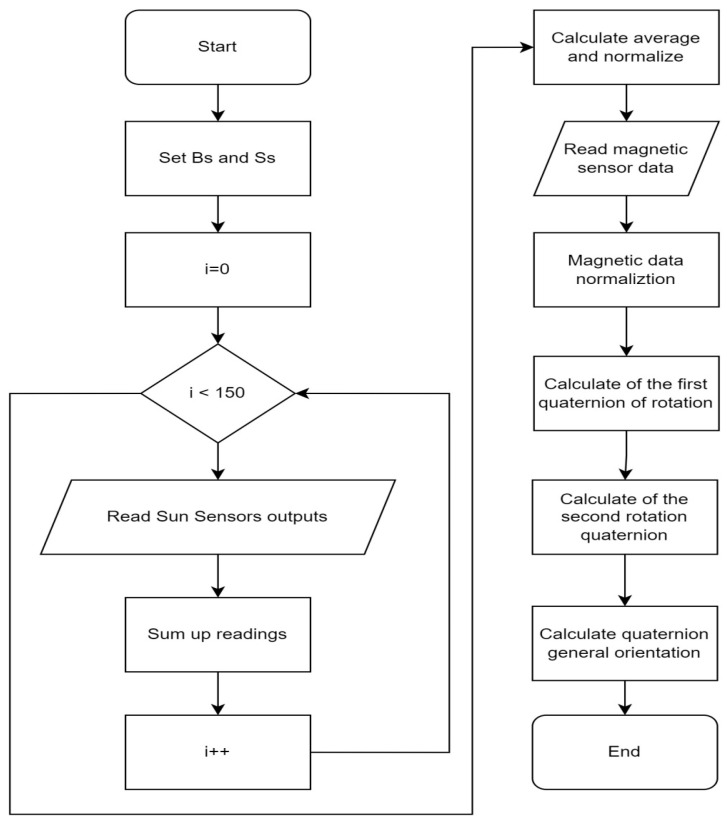
Orientation system computation algorithm.

**Figure 3 sensors-23-06388-f003:**
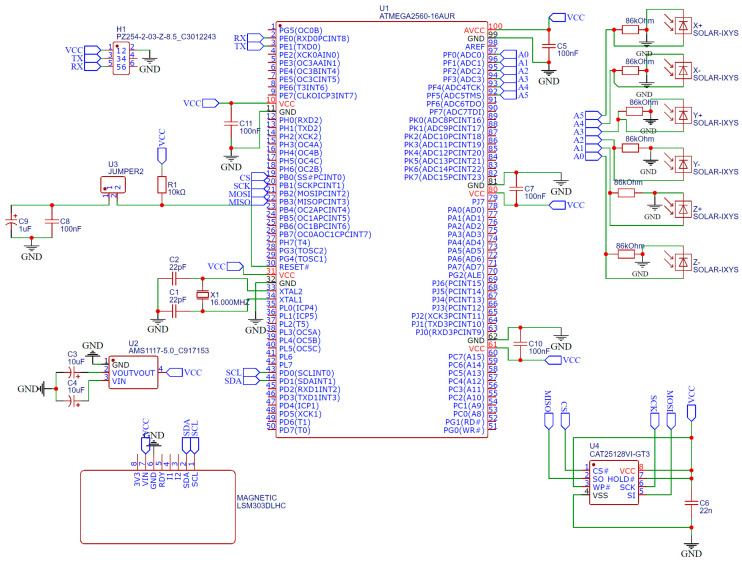
Electrical circuit of the orientation system.

**Figure 4 sensors-23-06388-f004:**
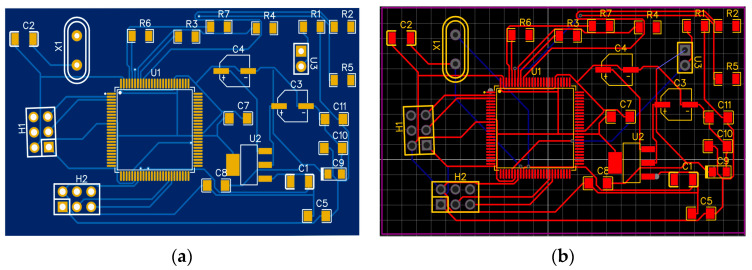
Parts setup diagram for making a double-sided circuit board. (**a**) First side of the board; (**b**) second side of the board.

**Figure 5 sensors-23-06388-f005:**
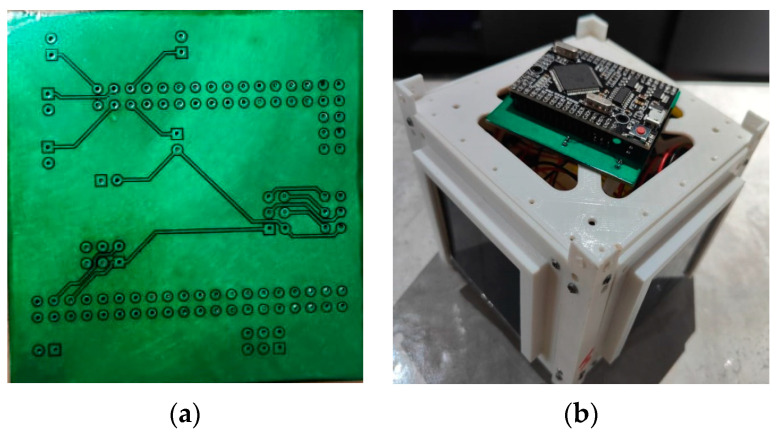
The board is completed using a CNC machine. (**a**) Printed circuit board; (**b**) ready-made board with a microcontroller.

**Figure 6 sensors-23-06388-f006:**
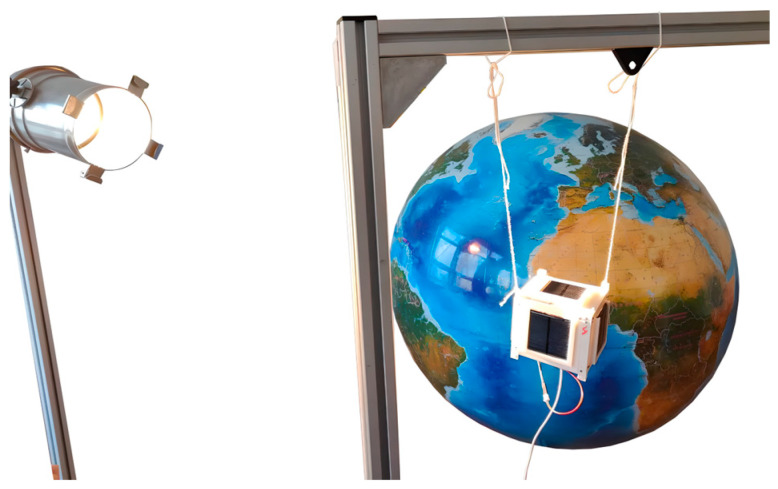
Experimental study using Cubesat.

**Figure 7 sensors-23-06388-f007:**
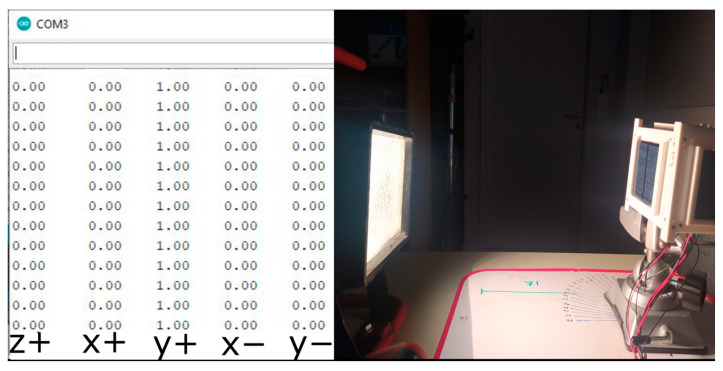
Measurement of the cosine of the angle on one side of the CubeSat.

**Figure 8 sensors-23-06388-f008:**
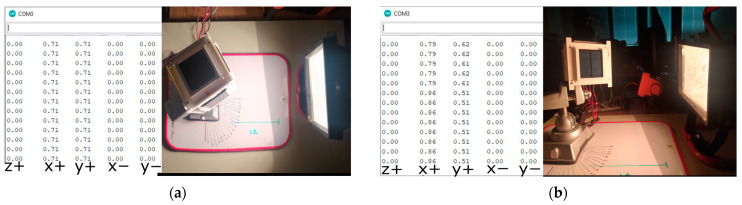
The fall of a sunbeam on two sides of a satellite. (**a**) The same angles of incidence of the beam; (**b**) different angles of incidence of the beam.

**Figure 9 sensors-23-06388-f009:**
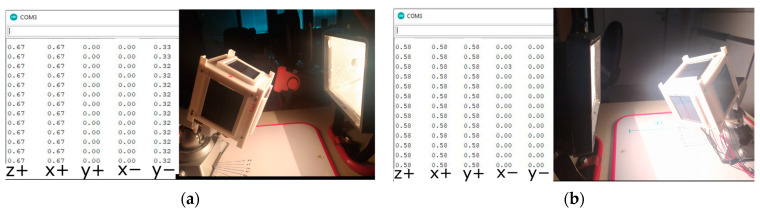
The fall of a sunbeam on three sides of a satellite. (**a**) The same angles of incidence of the beam; (**b**) different angles of incidence of the beam.

**Figure 10 sensors-23-06388-f010:**
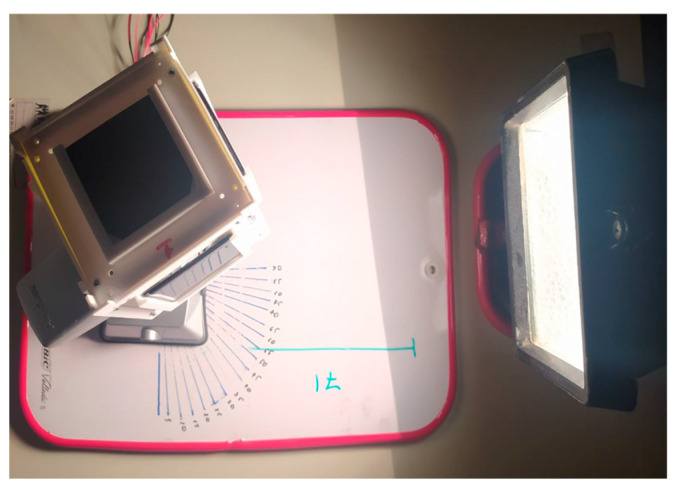
1 experiment. The Sun’s rays fall on only two faces of the Cubesat.

**Figure 11 sensors-23-06388-f011:**
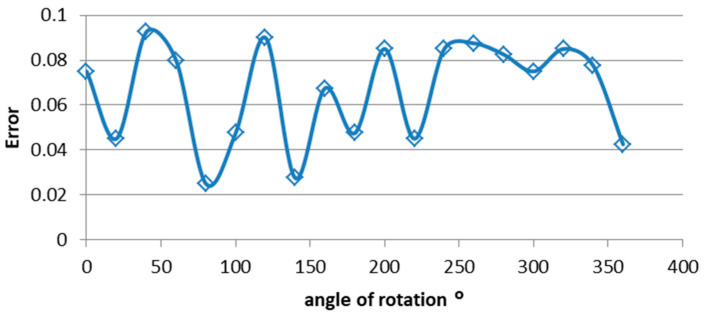
Sensor measurement error.

**Figure 12 sensors-23-06388-f012:**
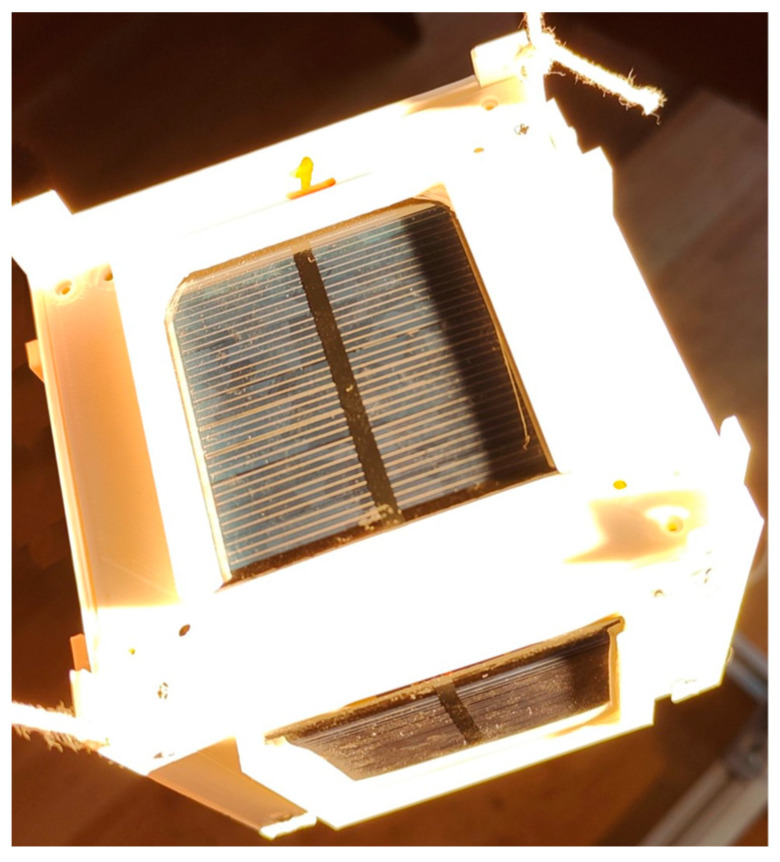
Experiment 2. The Sun’s rays fall on the three faces of the CubeSat.

**Table 1 sensors-23-06388-t001:** Satellite orientation data for two panels.

Angle of Rotation	Quaternion		Angle of Rotation	Quaternion	
0	Exp.	0.95, −0.17, −0.07, −0.01	200	Exp.	−0.22, −0.28, −0.09, 0.98
	Th.	1, 0, 0, 0		Th.	−0.17, 0.0, 0.0, 0.98
20	Exp.	0.94, −0.35, 0.01, −0.07	220	Exp.	−0.29, −0.23, −0.07, 1.00
	Th.	0.98, 0.0, 0.0, 0.17		Th.	−0.34, 0.0, 0.0, 0.93
40	Exp.	0.92, −0.39, 0.02, −0.35	240	Exp.	−0.50, −0.25, −0.11, 0.89
	Th.	0.93, 0.0, 0.0, 0.34		Th.	−0.49, 0.0, 0.0, 0.86
60	Exp.	0.89, −0.42, 0.02, −0.54	260	Exp.	−0.65, −0.25, −0.12, 0.79
	Th.	0.86, 0.0, 0.0, 0.49		Th.	−0.64, 0.0, 0.0, 0.76
80	Exp.	0.76, −0.24, 0.01, 0.77	280	Exp.	−0.74, −0.26, −0.13, 0.68
	Th.	0.76, 0.0, 0.0, 0.64		Th.	−0.76, 0.0, 0.0, 0.64
100	Exp.	−0.50, −0.18, −0.08, 0.97	300	Exp.	−0.84, −0.26, −0.14, 0.58
	Th.	0.64, 0.0, 0.0, 0.76		Th.	−0.86, 0.0, 0.0, 0.50
120	Exp.	0.31, −0.20, −0.09, 0.88	320	Exp.	−0.91, −0.26, −0.13, 0.44
	Th.	0.50, 0.0, 0.0, 0.86		Th.	−0.93, 0.0, 0.0, 0.34
140	Exp.	0.35, −0.22, 0.08, 0.95	340	Exp.	−0.96, −0.26, −0.14, 0.24
	Th.	0.34, 0.0, 0.0, 0.93		Th.	−0.98, 0.0, 0.0, 0.17
160	Exp.	0.22, −0.26, −0.07, 0.99	360	Exp.	−0.98, −0.20, −0.09, 0.11
	Th.	0.17, 0.0, 0.0, 0.98		Th.	−0.99, 0.0, 0.0, 0.0
180	Exp.	0.02, −0.27, −0.08, 0.99			
	Th.	0.0, 0.0, 0.0, 0.99			

**Table 2 sensors-23-06388-t002:** Satellite orientation data for three panels.

CubeSat Facets	Quaternion
+z, +x, +y (+z: 0.54 +x: 0.60 +y: 0.58 −x: 0.12 −y: 0.07)	−0.34, −0.36, −0.11, −0.99
+z, +x, −y (+z: 0.61 +x: 0.57 +y: 0.08 −x: 0.08 −y: 0.56)	−0.52, 0.14, 0.02, −0.99
+z, −x, −y (+z: 0.53 +x: 0.08 +y: 0.08 −x: 0.61 −y: 0.57)	0.66, −0.15, 0.43, −0.89
+z, +y, −x (+z: 0.60 +x: 0.05 +y: 0.60 −x: 0.52 −y: 0.01)	−0.25, −0.50, 0.23, −0.84

## Data Availability

The data that has been used is confidential.

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
