# Peer review of "Low-Cost Orientation Determination System for CubeSat Based Solely on Solar and Magnetic Sensors"

_sensors, 2023, doi:10.3390/s23146388_

Round 1

Reviewer 1 Report

The authors presented an interesting paper. The Paper is well-written and can be useful for practice. The research design, questions, hypotheses, and methods are clearly stated. The paper is suitable for the journal Sensors.
Minor review:
1. Please ensure the abstract is short but reflects the approach, results, and conclusions correctly and concisely.
2. Please check the keywords to ensure they are appropriate and complete.
3. All variables used in the equations should be clearly explained in the text. Similarly, formatting symbols in equations should be straightforward in the text. Please check SI units for all physical quantities.
4. I suggest the authors supplement the pictures with a more detailed description (Figures 7, 8, 9,10).
5. I suggest the authors edit Figure 12 (no border, vertical grid...) according to the magazine's template.
6. The references and the citation don't work the way the authors just added to the text.
7. In references, the DOI should be added for individual sources if possible.
8. In discussion, authors should discuss the results and how they can be interpreted from the perspective of previous studies and the working hypotheses. Future research directions may also be highlighted.

I suggest accepting the paper, but my comments should be resolved.

Author Response

1. Please ensure the abstract is short but reflects the approach, results, and conclusions correctly and concisely.

Response: 

CubeSats require accurate determination of their orientation relative to the Sun, Earth, and other celestial bodies to operate successfully and collect scientific data. This paper presents an orientation system based on solar and magnetic sensors that offers a cost-effective and reliable solution for CubeSat navigation. Solar sensors analyze the illumination on each face to measure the satellite's orientation relative to the Sun, while magnetic sensors determine the Earth's magnetic field vector in the satellite's reference frame. By combining the measured data with the known ephemeris of the satellite, the satellite-Sun vector and the magnetic field orientation can be reconstructed. The orientation is expressed using quaternions, representing the rotation from the internal reference system of the satellite to the selected reference system. The proposed system demonstrates the ability to accurately determine the orientation of a CubeSat using only two sensors, making it suitable for installations where more complex and expensive instruments are impractical. Additionally, the paper presents a mathematical model of a low-cost CubeSat orientation system and a hardware implementation of the sensor. The technology, using solar and magnetic sensors, provides a reliable and affordable solution for CubeSat navigation, supporting the increasing sophistication of miniature payloads and enabling accurate satellite positioning in space missions.

2. Please check the keywords to ensure they are appropriate and complete.

Response: CubeSat, satellite orientation, solar sensors, magnetic sensors, attitude determination, satellite-sun vector, magnetic field orientation, quaternions.
3. All variables used in the equations should be clearly explained in the text. Similarly, formatting symbols in equations should be straightforward in the text. Please check SI units for all physical quantities.

Response: all values in the equations in the text 
4. I suggest the authors supplement the pictures with a more detailed description (Figures 7, 8, 9,10).

Response: The drawings have been updated with better quality and more detailed descriptions are written
5. I suggest the authors edit Figure 12 (no border, vertical grid...) according to the magazine's template.

Response: Figure 12 corrected
6. The references and the citation don't work the way the authors just added to the text.

Response: All links are corrected and working
7. In references, the DOI should be added for individual sources if possible.

Response: All links are corrected and working
8. In discussion, authors should discuss the results and how they can be interpreted from the perspective of previous studies and the working hypotheses. Future research directions may also be highlighted.

A detailed description and future application of the system has been added to the discussion

All of the above comments have been addressed and highlighted in red in the attached document below. 

Reviewer 2 Report

I have reviewed the manuscript entitled " Low-cost orientation determination system for CubeSat based solely on solar and magnetic sensors " carefully. The authors introduce a cost-effective mathematical model for a CubeSat orientation system and describe the practical implementation of the sensor hardware. This system utilizes magnetic and solar sensors to determine the orientation of the CubeSat. To validate its effectiveness, a comprehensive laboratory experiment was conducted, and the paper provides a thorough explanation of the methodology employed. The proposed sensor represents orientation data using quaternions and incorporates elements of self-calibration for improved accuracy and reliability. The idea is interesting and deserves to be investigated. I found the calculations seem correct; the logic is convincing. However, I still have some possibly minor concerns:

1.     Abstract: CubeSat have become a popular tool……Have should be has, or CubeSat should be CubeSats.

2.     Fig. 8, Fig. 9, Fig. 10, and Fig. 11 are blurry. Do you have any clearer ones?

3.     The captions of the figures are too short. Can you add more information to the captions?

4.     Some of the sentences look like titles, please clarify. For example Page 4, line 161.

Add more information to the captions and I highly recommend extensive editing.

Author Response

  1. Abstract: CubeSat have become a popular tool……Have should be has, or CubeSat should be CubeSats.

Response: CubeSats require accurate determination of their orientation relative to the Sun, Earth, and other celestial bodies to operate successfully and collect scientific data. This paper presents an orientation system based on solar and magnetic sensors that offers a cost-effective and reliable solution for CubeSat navigation. Solar sensors analyze the illumination on each face to measure the satellite's orientation relative to the Sun, while magnetic sensors determine the Earth's magnetic field vector in the satellite's reference frame. By combining the measured data with the known ephemeris of the satellite, the satellite-Sun vector and the magnetic field orientation can be reconstructed. The orientation is expressed using quaternions, representing the rotation from the internal reference system of the satellite to the selected reference system. The proposed system demonstrates the ability to accurately determine the orientation of a CubeSat using only two sensors, making it suitable for installations where more complex and expensive instruments are impractical. Additionally, the paper presents a mathematical model of a low-cost CubeSat orientation system and a hardware implementation of the sensor. The technology, using solar and magnetic sensors, provides a reliable and affordable solution for CubeSat navigation, supporting the increasing sophistication of miniature payloads and enabling accurate satellite positioning in space missions.

  1. Fig. 8, Fig. 9, Fig. 10, and Fig. 11 are blurry. Do you have any clearer ones?

Response: Photos have been replaced by better quality

  1. The captions of the figures are too short. Can you add more information to the captions?

Response: Photo captions added

  1. Some of the sentences look like titles, please clarify. For example Page 4, line 161.

Response: Sentence corrected

All your comments are taken into account and corrected. The document below is highlighted in red.

Reviewer 3 Report

This paper deals with the ¨Low-cost orientation determination system for CubeSat based solely on solar and magnetic sensors¨. The manuscript topic is interesting, but a couple of issues can be detected in this article that needs to be majorly revised:

1- The abstract is very short and confusing. It is hard to follow the author's approach in their research. Please add the exact methodology and more quantities of results.

2- A lot of typos and grammatical errors can be detected in the manuscript.

3- The aim and scope of this paper are not clearly described. The author should state what kind of problem want to solve in this research.

4- The introduction is brief and not sufficient to understand the state of the authors.

5- Please remove bunch citing, like [1-5] or [14-16]. If the pieces of literature are important, please separately explain their results and output.

6- the equations need references.

7- The quality of images is low. Replace them with high-resolution images.

8- Please replace the results in Figs—8 and 9 with Tables.

9- Sections 2, 3, and 4 are almost the experimental procedure and must combine.

10- There is almost no discussion of results, and it is the main weakness of the paper. Please improve comprehensively the discussion on the results presented in Section 6.

11- The writing structure of the paper is not matched by a scientific paper. Please follow the ¨Instruction of Authors¨ and use the journal template.

Mentioned earlier

Author Response

1- The abstract is very short and confusing. It is hard to follow the author's approach in their research. Please add the exact methodology and more quantities of results.

Response: CubeSats require accurate determination of their orientation relative to the Sun, Earth, and other celestial bodies to operate successfully and collect scientific data. This paper presents an orientation system based on solar and magnetic sensors that offers a cost-effective and reliable solution for CubeSat navigation. Solar sensors analyze the illumination on each face to measure the satellite's orientation relative to the Sun, while magnetic sensors determine the Earth's magnetic field vector in the satellite's reference frame. By combining the measured data with the known ephemeris of the satellite, the satellite-Sun vector and the magnetic field orientation can be reconstructed. The orientation is expressed using quaternions, representing the rotation from the internal reference system of the satellite to the selected reference system. The proposed system demonstrates the ability to accurately determine the orientation of a CubeSat using only two sensors, making it suitable for installations where more complex and expensive instruments are impractical. Additionally, the paper presents a mathematical model of a low-cost CubeSat orientation system and a hardware implementation of the sensor. The technology, using solar and magnetic sensors, provides a reliable and affordable solution for CubeSat navigation, supporting the increasing sophistication of miniature payloads and enabling accurate satellite positioning in space missions.

2- A lot of typos and grammatical errors can be detected in the manuscript.

Response: All text is corrected

3- The aim and scope of this paper are not clearly described. The author should state what kind of problem want to solve in this research.

Response:  One of the main goals of the SAT mission is to develop a CubeSat orientation algorithm that will exceed a few degrees of accuracy when using two sensors, particularly in sunlight. This objective aims to overcome the limitations of using complex and expensive tools like star sensors. Consequently, this paper presents an orientation system based solely on two sensors: a solar sensor and a magnetometer, which can be easily installed on any CubeSat. This system offers a practical alternative for situations where more elaborate instruments are not feasible.

Still, the utilization of solar and magnetic sensors in CubeSat orientation systems provides several advantages. Firstly, it is a cost-effective and reliable solution, making it accessible to a wide range of CubeSat missions. Secondly, the availability of solar and Earth's magnetic fields throughout most regions on our planet enables the utilization of this technology for CubeSat navigation, regardless of the mission's location.

4- The introduction is brief and not sufficient to understand the state of the authors.

Response:  

An important aspect of CubeSat operations is determining the satellite's orientation relative to the Sun, Earth, and other celestial bodies. One technology used in the development of CubeSat orientation sensors is the use of solar and magnetic sensors [1,2]. This information is necessary for spacecraft orientation [3,4] control and collecting accurate scientific data [5]. The purpose of the solar sensor is to determine the coordinates of the "satellite-sun" vector [6,7] in the reference frame associated with the satellite. This is achieved by processing photoelectric signals from solar sensors located on the six sides of the satellite [8,9].

Modeling shows that the precision of attitude determination can be better than 1º. Furthermore, the Radio Aurora Explorer 3U CubeSat [10] demonstrated back in 2014 that it is possible to obtain information about the satellite's position with accuracy better than 1º (solar light) in orbit [11,12]. This is especially true when the satellite is on the sunny side, as multiple sensors are used for successful task execution. However, this determination is more difficult during eclipses [13]. Determining the orientation of CubeSat satellites remains a challenging task, as they are small and still lack active orientation and control systems [14, 15]. Additionally, new miniature payloads for space-based missions are becoming increasingly complex and require accurate knowledge of the satellite's position. One of the objectives of the SAT mission is to develop a CubeSat orientation algorithm using two sensors that exceeds a few degrees in accuracy in sunlight.

One of the main goals of the SAT mission is to develop a CubeSat orientation algorithm that will exceed a few degrees of accuracy when using two sensors, particularly in sunlight. This objective aims to overcome the limitations of using complex and expensive tools like star sensors. Consequently, this paper presents an orientation system based solely on two sensors: a solar sensor and a magnetometer, which can be easily installed on any CubeSat. This system offers a practical alternative for situations where more elaborate instruments are not feasible.

Still, the utilization of solar and magnetic sensors in CubeSat orientation systems provides several advantages. Firstly, it is a cost-effective and reliable solution, making it accessible to a wide range of CubeSat missions. Secondly, the availability of solar and Earth's magnetic fields throughout most regions on our planet enables the utilization of this technology for CubeSat navigation, regardless of the mission's location.

In summary, this paper aims to present a comprehensive orientation system for CubeSats based on solar and magnetic sensors, highlighting its feasibility and advantages. The proposed system, utilizing only two sensors, offers a practical solution for determining CubeSat orientation and addresses the challenges posed by limited hardware and energy resources. Through its cost-effectiveness, reliability, and accessibility, the technology opens up new possibilities for CubeSat missions, supporting their growing complexity and enabling precise knowledge of the satellite's position in space-based operations.

The scope of this orientation system based on solar and magnetic sensors extends to CubeSat operations and scientific data collection, also encompasses the development of a mathematical model, hardware implementation, and practical application of the orientation system.

The novelty of this work lies in the mathematical model, which is a new methodology for more accurate determination of the orientation of nanosatellites using budget sensors.

The article is structured as follows: Section 2 describes the mathematical model of the satellite orientation system and presents the sensor architecture. Section 3 discusses the operational algorithm and the hardware implementation of the sensor. In Section 4, a detailed presentation of the results obtained from studying the algorithm of the proposed system in laboratory conditions is provided. The concluding remarks are presented in Section 5.

5- Please remove bunch citing, like [1-5] or [14-16]. If the pieces of literature are important, please separately explain their results and output.

Response: An important aspect of CubeSat operations is determining the satellite's orientation relative to the Sun, Earth, and other celestial bodies. One technology used in the development of CubeSat orientation sensors is the use of solar and magnetic sensors [1,2]. This information is necessary for spacecraft orientation [3,4] control and collecting accurate scientific data [5]. 

Determining the orientation of CubeSat satellites remains a challenging task, as they are small and still lack active orientation and control systems [14, 15].

6- the equations need references.

Response: All equations are referenced

7- The quality of images is low. Replace them with high-resolution images.

Response: The images have been replaced

8- Please replace the results in Figs—8 and 9 with Tables.

Response: These pictures cannot be replaced in tabular form, as we show how the sun's beam is fed to the satellite and the data from the sensors. These photos have been replaced with better quality photos. 

9- Sections 2, 3, and 4 are almost the experimental procedure and must combine.

Response:

The chapters are combined into one chapter. 

10- There is almost no discussion of results, and it is the main weakness of the paper. Please improve comprehensively the discussion on the results presented in Section 6.

Response: 

This paper includes a discussion of the results obtained from laboratory tests conducted on an inexpensive CubeSat in the CubeSat format. The primary objective of these tests was to determine the coordinates of the "Satellite-Sun" vector in the satellite's reference frame using magnetic sensors and sun sensors. These coordinates were then utilized to accurately determine the CubeSat's orientation. By incorporating a magnetic sensor and developing a mathematical model of the CubeSat's passive orientation system, experimental testing was conducted under controlled laboratory conditions. The results of these tests demonstrated the system's capability to accurately determine the satellite's orientation throughout the orbital day.

In the first experiment, the proposed sensor of the passive orientation system of the nanosatellite was tested. A comprehensive analysis of the solar panels was performed, and a mathematical model for the solar sensors was constructed. Additionally, experiments were conducted at various distances from the solar simulator. The detailed data and findings of this experiment can be found in the source provided [16].

In the second experiment, aimed at enhancing the accuracy of the nanosatellite's orientation, a second sensor was incorporated to measure the Earth's magnetic field. Following the integration of this second sensor, a mathematical model of the complete passive orientation system was developed. The outcome of this experiment yielded an orientation sensor with an error rate of 1.2%, which is a significant achievement in laboratory conditions.

In conclusion, the exclusive utilization of solar and magnetic sensors for orientation determination presents a cost-effective solution for CubeSats. These sensors are generally affordable and have low power requirements, making them particularly advantageous for CubeSats with limited resources. Moreover, the combination of information gathered from these sensors enables accurate orientation determination without the need for additional sensors or complex algorithms. This straightforward and cost-efficient method of orientation determination represents a noteworthy advancement in enhancing accessibility to spaceflight for a broader range of organizations.

It is worth noting that this system exhibits potential for further expansion in the future. It can be adapted to accommodate various installations, incorporate different sensor types, facilitate scientific research, and investigate different anomalies. By harnessing the benefits of solar and magnetic sensors, CubeSat missions can continue to evolve and effectively address the increasing demands of space exploration and scientific investigations.

In summary, the results obtained from the laboratory tests validate the effectiveness and reliability of the proposed orientation system based on solar and magnetic sensors. These results instill confidence in the system's ability to accurately determine the CubeSat's orientation, thereby supporting the successful operation and data collection of CubeSat missions. With its affordability, simplicity, and adaptability, this orientation system offers a promising avenue for advancements in CubeSat technology and expands the opportunities for organizations to participate in space-based activities.

11- The writing structure of the paper is not matched by a scientific paper. Please follow the ¨Instruction of Authors¨ and use the journal template.

Response: The article has been corrected to the standard of the magazine. All comments have been taken into account. Please check the article in its entirety, since the text has been updated by 30%. 

Reviewer 4 Report

Put more actual papers from MDPI Journals and IEEE Journals and Conferences in the references list.

Explain in more details the electrical circuit of the orientation system.

I did not notice any grammatical or syntax errors.

Author Response

Please see the attachments, as the text of the article has been changed by 30%. 

Reviewer 5 Report

This paper is about the implementation of lab tests of CubeSat. To get the orientation of the CubeSat, magnetic and sun sensors are employed.Also a mathematical model is developed with the help of magnetic field vector. These sensors are inexpensive and require less power according to the authors. The language of quaternions introduced by Hamilton is very helpful in navigational aids. The mathematical notation is very compact. This paper is beautiful also because, a hardware implementation using microcontrollers is shown. Now this can be used as a prototye and more advanced systems can be designed.     

English is fine. One might make minor changes for making it better, but no issues.

Author Response

(The authors gave the same response as above.)

Round 2

Reviewer 3 Report

The authors answered the comments properly.